# Preparation and Properties of Low-Carbon Foamed Lightweight Soil with High Resistance to Sulphate Erosion Environments

**DOI:** 10.3390/ma16134604

**Published:** 2023-06-26

**Authors:** Yongsheng Wang, Huiwen Wan, Hao Liu, Gaoke Zhang, Xiaoyang Xu, Cong Shen

**Affiliations:** 1China Construction Second Engineering Bureau Limited East China Branch, Shanghai 200135, China; wangyongsheng_@cscec.com (Y.W.); liuhao-2@cscec.com (H.L.); xuxiaoyang_@cscec.com (X.X.); 2School of Materials Science and Engineering, Wuhan University of Technology, Wuhan 430070, China; 256935@whut.edu.cn; 3School of Resources and Environmental Engineering, Wuhan University of Technology, Wuhan 430070, China; gkzhang@whut.edu.cn

**Keywords:** foamed lightweight soil, road base, sulphate attack resistance, carbon emissions

## Abstract

Foamed lightweight soil (FLS) is a lightweight cementitious material containing a large number of tiny closed pores and has been widely used as a filler in places such as railways, roads and airports. However, there has been little research into the resistance of FLS to sulphate attack in practical engineering applications. The performance of FLS against different sulphate erosion concentrations was studied to elucidate the engineering characteristics of using large volumes of FLS as fill material for the road base in the construction of intelligent networked vehicle test sites. The results showed that the compressive strength of FLS prepared using 30% Portland cement (C), 30% granulated blast furnace slag (GBFS), 40% fly ash (FA) and a small amount of a concrete antiseptic agent (CA) as cementitious materials reached 0.8 and 1.9 MPa at 7 and 28 d, respectively, when the wet density was about 600 kg/m^3^, which met the design requirements. The FLS prepared via the above-mentioned cementitious system had a low carbon emission, with a CO_2_ emission reduction rate of up to 70%. It also had excellent sulphate attack resistance: the corrosion resistance coefficient of the cementitious material system reached 0.97, which was considerably better than that of C (0.83). For an erosion medium environment with SO_4_^2−^ concentrations of less than 1000 mg/L (moderate), 40% GBFS or FA can be used to prepare FLS. When the concentration of SO_4_^2−^ is less than 4000 mg/L (severe), 30% C, 30% GBFS and 40% FA can be used as cementitious materials, preferably in combination with an appropriate amount of CA, to prepare FLS.

## 1. Introduction

In road traffic engineering, uneven settlement is often induced by poorly compacted or improperly backfilled road foundations, especially in highly filled sections and soft base sections, which, in turn, seriously affects the quality of road construction [1,2]. In recent years, foamed lightweight soil (FLS) has replaced conventional fill and has been widely used in road base treatment, the widening of existing roads, the filling of soft foundation bridge abutments and the filling of steep sections in mountainous areas [3,4]. Compared with conventional fill, FLS has significant advantages, such as light weight, higher fluidity, easy application, high strength after hardening, good integrity, higher durability and low cost. These advantages greatly reduce the load, soil pressure and thickness of the shallow layer in road base treatment, thereby improving the quality of the road base construction [5,6,7,8].

FLS is a lightweight material containing a large number of small closed pores. These pores are formed by turning a foaming agent aqueous solution into foam, mixing it evenly with cement slurry in a certain proportion and pumping the mixture into the construction site for natural curing [9]. The compressive strength of FLS is usually between 1.0 and 1.5 MPa at 28 d, with a wet density of 600 kg/m^3^, according to general design specifications. The cementitious materials used during the preparation of FLS usually contain large amounts of fly ash (FA), granulated blast furnace slag (GBFS) or other supplementary cementitious materials in place of Portland cement (C) [10,11].

Intelligent networked vehicle testing sites are large road-based projects that have been widely built in recent years. They cover large areas and are generally constructed around cities, away from downtown areas and usually near seas or lakes. The soil and groundwater at such sites often contain high concentrations of sulphate. During the construction of intelligent networked vehicle testing sites, FLS with depths of several metres is usually used as basement fill, resulting in the usage of up to one million square metres of FLS (Figure 1). GBFS and FA are usually used to replace C during the preparation of FLS to improve the durability of FLS in high-sulphate environments and reduce costs [12].

The replacement of ordinary fill with FLS in construction has been extensively studied. In a previous study on the application of FLS in the road base during highway widening, FLS exhibited a higher compressive strength, better water resistance and less settlement compared with ordinary fill [13,14,15]. Another study on the use of FLS during road widening showed that the settlement of the test section filled with FLS was much less than the allowable settlement value, indicating that FLS is a highly suitable road fill material in road expansion projects [16]. A study on the application of FLS in soft soil roadbeds for high-speed railways indicated similar results [17]. A study conducted by Yoichi Watabe and Takatoshi Noguchi on the relationship between vertical displacement and age after filling the runway of a Tokyo airport with FLS concluded that FLS is an excellent, economically beneficial fill material for airport runway roadbeds [18]. Moreover, many studies have shown that FLS is suitable for reducing the settlement of soft ground and maintaining the stability of embankments [19,20].

The effects of FLS composition and microstructure on its properties have been investigated by many researchers as well. Panesar examined the effect of composite and protein foaming agents on the performance of FLS and concluded that a composite foaming agent leads to better FLS performance [21]. Several studies on the effects of supplementary cementitious materials on the workability and mechanical properties of FLS indicated that the addition of silica fume and FA can improve the compressive strength and fluidity of FLS [22,23,24]. Paul J. Tikalshy and other researchers found that the depth of penetration is the key factor affecting the freeze–thaw resistance of FLS [25]. M.R. Jones and A. McCarthy compared the properties of FLS prepared with FA and FLS made with sand [26]. The results showed that the FLS with FA had better performance, especially in terms of fluidity, compressive strength and sulphate attack resistance.

Due to the late start of research on FLS in China, large-scale promotion and application are yet to be popularised, while FLS is generally not used for structural support and does not need to be studied for properties such as dry density, water absorption or thermal conductivity like foam concrete [27,28,29]. Many projects, depending on the geological soil conditions of the construction site and the surrounding service environment, require real-time targeted measures for the FLS in service [30,31]. The FLS used in the projects on which this research is based amount to more than 680,000 and are mostly built on soft ground such as fishing ponds and river embankments, although there are more than 60,000 infill piles in the lower part. The quality and durability of the foam lightweight soils play a very important role in the construction quality of the test site road foundations. However, there are currently few requirements for durability performance indicators of FLS in Chinese technical specifications or standards. Most of the research reports and thesis literature are related to the study of the entry resistance of foam lightweight soil or foam concrete.

In this study, the effects of the use of GBFS, FA and CA on the compressive strength, sulphate resistance and carbon footprint of FLS were characterised to evaluate the feasibility of using FLS as the basement fill of intelligent networked vehicle testing sites near coasts, salt lake areas and inland lakes with groundwater containing high concentrations of sulphate ions. It is believed that the research in this paper can provide important guidance and reference value for the future construction and engineering of FLS in high-sulphate environments.

## 2. Materials and Methods

### 2.1. Materials

The main raw materials use in this study were C, FA, GBFS, CA and the foaming agent.

The C is PO 42.5 cement produced by China Conch Cement Co. Ltd. (Wuhu, China); its chemical composition and basic properties are shown in Table 1 and Table 2, respectively. GBFS is S95-grade ground slag produced by Hubei Jinshenglan Metallurgical Technology Co. Ltd. (Xianning, China); it has a density of 2800 kg/m^3^ and a specific surface area of 410 m^2^/kg, and its chemical composition is presented in Table 1. FA was Class F Grade II FA produced by Hubei Koneng Environmental Protection Co., Ltd. (Hanchuan Power Plant, Hanchuan, China); its density is 2200 kg/m^3^, its fineness (through a 45 µm square-hole sieve) is 15% and its chemical composition is shown in Table 1. The CA was produced by Hubei Granular Solid Silica Fume Co., Ltd. (Wuhan, China); its main components are metakaolin and silica fume, its density is 2100 kg/m^3^ and its main chemical composition is presented in Table 1. The particle size distribution for the raw materials is shown in Figure 2.

The foaming agent is a ready-mixed composite foaming agent, model JY-SRN2, produced by the Guangdong Shengrui Technology Company (Guangzhou, China). 

### 2.2. Mix Design

According to the design requirements of FLS as a road base fill material, the technical specifications of the utilised FLS were as follows: freshly mixed FLS flow factor (flow degree) of 170 ± 10 mm, wet density of 600 ± 30 kg/m^3^, 7 d compressive strength ≥0.5 MPa and 28 d compressive strength ≥1.0 MPa. The FLS material composition and combinations are shown in Table 3. Group A had C only, group B had C and GBFS, group C had C and FA, group D had C, GBFS and FA, and group E had C, GBFS, FA and CA (5% of the mass of C).

The water–binder ratio was 0.65. Every per cubic FLS contains the volume of the cementitious material and water, with the remainder being the volume of the foam cluster.

### 2.3. Specimen Preparation

#### 2.3.1. Fabrication of Corrosion Resistance Specimens for Gelling Material Systems

According to GB/T749-2008 [32] (the test method of determining the capability of cement to resist sulphate erosion), the cementitious material system consisted of C, GBFS, FA and CA. The mass ratio of cementitious material to standard sand was 1:2.5, and the water–cement ratio was 0.5. The test mould was made of 10 mm × 10 mm × 60 mm prismatic specimens, and six specimens were formed in each group. The formed specimens were removed from their moulds at 24 h and then placed in a humid heat maintenance box at 50 °C for 7 d. In each group, three specimens were immersed in an erosion solution for 28 d (3% sodium sulphate solution; 20 °C ambient temperature), and the three remaining specimens were placed in water at 20 °C for 28 d.

#### 2.3.2. Fabrication of FLS Cube Specimens

First, the foaming agent was diluted with water in a ratio of 1:99 to 100 times the original. An intelligent mini foaming machine was turned on, and its knob was adjusted to obtain a foam density of 50 g/L. According to the material ratio in Table 4, a certain amount of cementitious material was mixed into a slurry in a water–binder ratio of 0.65, and this slurry was mixed with the corresponding foam cluster to form FLS. The FLS flow factor and wet density were tested. The mixture was poured into a 100 mm × 100 mm × 100 mm mould (one mould contains three cubic specimens) such that the height of the slurry was slightly higher than the test mould. The surface was covered with cling film and let sit at a temperature of 20 °C and humidity of 95% for maintenance (standard maintenance conditions). After 8 h, the slurry above the mould was scraped off with a scraper, the cling film was replaced and maintenance was continued until 48 h, when the samples were demoulded. The demoulded specimens were bagged in plastic and returned to the above conditioning environment until the appropriate days. The main preparation process is shown in Figure 3.

### 2.4. Test Methods

#### 2.4.1. Corrosion Resistance Coefficients of Cementitious Materials

The flexural strength *R*_1_ of the specimens immersed in the 3% sodium sulphate solution for 28 d and the flexural strength *R*_0_ of the specimens maintained in water at 20 °C for the same age were tested separately. The coefficient of resistance *K* of the specimens was
(1)$$K=\frac{R_{1}}{R_{0}}$$
where *K* is the coefficient of resistance to corrosion, *R*_1_ is the flexural strength of the specimens immersed in sulphate for 28 d (MPa) and *R*_0_ is the flexural strength of the specimens maintained in water at 20 °C for the same age (MPa).

#### 2.4.2. Flow Factor (Fluidity), Wet Density, Compressive Strength and Resistance to Sulphate Attacks of FLS

Flow factor of freshly mixed foam: A hollow cylinder with an inner diameter and height of 80 mm was placed on a glass plate, and the mixed foam was poured into the cylinder. The cylinder was tapped with the scraper as it was being filled with slurry, and the top was scraped flat. Then, the cylinder was quickly lifted vertically to allow the slurry to collapse naturally. It was left to stand for 1 min, and the slurry diameter was measured with a steel ruler along the maximum diameter direction and the direction perpendicular to it. The arithmetic mean was taken as the result of this flow factor test.

Wet density of FLS: A 1 L volumetric cylinder was placed on an electronic scale and cleared. Slurry was poured into the cylinder, the cylinder wall was tapped with the scraper as it was filled with slurry and excess slurry was scraped off with the scraper. The mass of the cylinder filled with slurry *M* was determined, and the wet density of FLS was
(2)$$\rho =\frac{M}{V}$$
where *ρ* is the wet density (kg/m^3^), *M* is the mass of the slurry filling the cylinder (g) and *V* is the volume of the volumetric cylinder (1 L).

Compressive strength of hardened FLS: After the cube block had cured and expired, it was removed from the curing room, and its side length *L* was measured with a dial calliper. It was then placed on a pressure-testing machine (TYE-300D) for pressing. The rate of pressing load was controlled at 0.1 ± 0.05 kN/s, and the test block breaking load *P* was recorded. The compressive strength of hardened FLS was
(3)$$F=\frac{P}{L^{2}}$$
where *F* is the compressive strength of the test block (MPa), *P* is the test block breaking load (N) and *L* is the side length of the test block (mm).

Sulphate attack resistance of FLS: The FLS specimens were maintained under standard conditions for 28 d. One group of comparison specimens was maintained under standard conditions for another 28 d. The other group was maintained in solutions with different SO_4_^2−^ concentrations for the same length of time. After the appropriate age of maintenance, the compressive strength of the specimens was measured. Solutions with different SO_4_^2−^ concentrations were prepared using a Na_2_SO_4_ chemical reagent with SO_4_^2−^ concentrations of less than 200 mg/L (mild), 200 to less than 1000 mg/L (moderate) and 1000 to less than 4000 mg/L (severe).

## 3. Results and Discussion

### 3.1. Resistance of Cementitious Materials to Sulphate Attacks

Table 4 shows the coefficients of resistance to sulphate attacks of the cementitious materials. Specimen A exhibited the worst resistance, which was mainly due to the large amount of Ca(OH)_2_ generated in the system. This large amount of Ca(OH)_2_ reacted with the erosion medium (Na_2_SO_4_) to produce CaSO_4_, which then reacted with hydrated calcium aluminate to produce AFt. This increased the volume of the solid phase and generated considerable crystallisation pressure, leading to a decrease in compressive strength and even swelling and cracking or damage. When 40% GFBS (specimen B) or 40% FA (specimen C) was used instead of C, the sulphate attack resistance was significantly improved due to the reduced amount of Ca(OH)_2_ generated in the system, but the effect of GBFS replacement was better than that of FA replacement. The amount of C in specimen D was only 30%, and the amount of GBFS and FA reached 70%. Although the compressive and flexural strengths of specimen D after curing in water were lower than those of specimens A, B and C, the strengths of specimen D after being immersed in the sulphate solution decreased less than those of the other specimens, especially specimen E (which contained a small amount of CA), whose mechanical properties decreased slightly. Thus, the replacement of C with FA and GBFS in the cementitious material system could enhance sulphate attack resistance. Moreover, the smaller the amount of CA mixed with FLS, the more significant the resistance of FLS to sulphate attacks.

### 3.2. Physical Properties of FLS

Some performances of the FLS specimens are shown in Table 5. The flow factor of the freshly mixed FLS was 165–178 mm, and the wet density was 593–608 kg/m^3^, which both met the design requirements. After they hardened, specimen A had the highest compressive strength, whereas specimen C had the lowest compressive strength. Specimen C had 7 and 28 d compressive strengths of 0.63 and 1.34 MPa, respectively, which fulfilled the design requirements (7 d compressive strength ≥0.5 MPa and 28 d compressive strength ≥1.0 MPa).

It should be emphasised that the quality of the FLS has a great relationship with the quality of the foam. The pH value of the foaming agent selected in this study is 10.2, which is alkaline and compatible with cement-based cementitious material systems. It not only has high dilution times, but also a good foaming effect and is stable for a long time.

From the point of view of practical engineering applications, it is better to adopt the material composition and proportioning of specimen D or E due to the small amount of C, which can reduce production costs and decrease the temperature rise inside the bulk FLS, thereby helping prevent the FLS surface from cracking.

### 3.3. Low Carbon Emission of FLS

For every 1 t of cement produced, about 0.7 t of CO_2_ is emitted into the atmosphere and about 100 kW/h of electricity is consumed. In accordance with the material composition and ratios of specimen A, at least 241.5 kg of CO_2_ would be emitted per cubic metre of FLS; specimen B or C, about 144.9 kg; and specimen D or E, only 72.5 kg, as shown in Table 6. Hence, for the same FLS that met the design requirements, specimen D or E could reduce CO_2_ emissions by 169 kg/m^3^ compared with specimen A (a reduction of 70%) while eliminating 200 kg of solid waste, such as slag and FA, and reducing raw material costs by more than 20%. For the construction of a large intelligent networked vehicle test site, a typical roadbed needs to be filled with at least 500,000 m^3^ of FLS, where at least 80,000 t of CO_2_ emissions can be eliminated by using the preparation of specimen D or E.

### 3.4. Resistance of FLS to Sulphate Attacks

The specimens were divided into two groups after being maintained under standard conditions to investigate the resistance of FLS to sulphate attacks. Over another 28 d, one group of comparison samples continued under standard conditions, whereas the other group was immersed in solutions with different SO_4_^2−^ concentrations. The changes in the compressive strengths of the specimens immersed in solutions with different SO_4_^2−^ concentrations are shown in Table 6.

As seen in Table 7, the compressive strengths of the specimens immersed in the 200 mg/L SO_4_^2−^ solution remained unchanged and, in some cases, even increased slightly compared with those of the specimens under standard curing conditions (56 d, temperature of 20 °C, humidity of 95%). The results were mainly determined by the FLS microstructure, as shown in Figure 4. Macroscopic spherical pores and hardened slurry accounted for 75–80% and 20–25% of the whole volume. These pores had varying diameters and were closed and not connected to each other. The pore wall wrapped around the pores (hardened slurry) was the main source of strength. Because the water–binder ratio was large (0.65), this part of the hardened slurry also contained micron-sized capillaries and nano-sized gel pores. It was found that the pore structures of sample A and B were poor, but their compressive strength of 56 d was higher. The pore size distribution of sample C and sample D did not differ much, but their compressive strengths differed significantly. The main reason for the above two phenomena was the different pore wall structures (hardened slurry).

Due to the low concentration of eroded SO_4_^2−^, less delayed AFt was formed in the hardened slurry. Consequently, the hardened slurry became denser and therefore appeared to increase rather than decrease in strength. The trend curves of the changes in the compressive strengths of all specimens after erosion by different SO_4_^2−^ concentrations are shown in Figure 5.

However, as the SO_4_^2−^ concentration increased, the compressive strengths of the specimens decreased. The compressive strength of specimen A decreased significantly when the SO_4_^2−^ concentration increased from 200 to 4000 mg/L. The compressive strength decreased by 11.9% and 23.7%, respectively, which were the largest decreases among all of the specimens, mainly because the Ca(OH)_2_ generated in C reacted with a large amount of SO_4_^2−^ to produce delayed AFt, which resulted in expansion and reduced the compressive strength. Had the immersion been prolonged further, the strength would have decreased even more and collapse could have occurred. When GBFS or FA was used to partially replace C, the compressive strengths of the specimens decreased as well. Nonetheless, the decrease tended to be significantly slower; the compressive strengths of specimens B and C in the 4000 mg/L solution decreased by 13.6% and 17.4%, respectively. Therefore, the admixture of GBFS or FA did improve resistance to sulphate corrosion. Under the same erosion condition (SO_4_^2−^ concentration of 4000 mg/L), the compressive strength of specimen D, which contained less C, GBFS and FA, decreased by only 7.5%. These experimental data showed that after sulphate attacking, the compressive strength of sample A decreased the most and sample D decreased the least, which was related to the pore size distribution as well as its own material composition. Specimen E’s compressive strength decreased by only 4.3%, but its resistance to sulphate erosion was more significant. The main reasons for the excellent resistance of specimen E to sulphate erosion were as follows: The low C dosage in the cementitious material system resulted in a small amount of corrosion-prone Ca(OH)_2_ generated by the reaction. By contrast, CA consisted of large amounts of partial kaolinite and silica fume; they contained large quantities of active SiO_2_ and Al_2_O_3_, which were conducive to the generation of hydrated calcium silicate (C–S–H). In addition, the specific surface areas of kaolinite and silica fume were large, and they filled the pores of the porous pore wall slurry, which was also beneficial for strength improvement.

The above analyses show that pure C should not be utilised as a cementitious material when using FLS as an underground fill material in seaside, salt lakes or groundwater containing high concentrations of sulphate ions. For an erosion medium environment with SO_4_^2−^ concentrations of less than 1000 mg/L (moderate), GBFS or FA can be used to replace part of C. For SO_4_^2−^ concentrations of less than 4000 mg/L (severe), less C should be used, and large quantities of GBFS and FA should be adopted as cementitious materials, preferably mixed with an appropriate amount of CA. At present, there is more research on the strength and durability of concrete predicted by machine learning [33,34,35,36,37]. It is believed that in future study, artificial intelligence algorithms will also have good prospects in the field of the FLS prediction of strength and durability.

## 4. Conclusions

(1)FLS prepared using 30% C, 30% GBFS, 40% FA and a small amount of CA as a cementitious material has excellent resistance to sulphate erosion, low carbon emission and a corrosion resistance coefficient reaching 0.97. Compared with FLS prepared using pure cement, it can reduce CO_2_ by 70% and has obvious low carbon characteristics.(2)FLS with a flow factor of about 170 mm and a wet density of about 600 kg/m^3^ can be prepared using 30% C, 30% GBFS, 40% FA, a water–binder ratio of 0.65 and a foam density of 50 kg/m^3^. Its compressive strengths at 7 and 28 d reached 0.8 and 1.9 MPa, respectively. After sulphate attacking, its compressive strength decreased the least, which is related to its pore size distribution and its own material composition.(3)For areas where the groundwater contains high concentrations of sulphate ions, pure C should not be utilised as a cementitious material to prepare FLS. For an erosion medium environment with SO_4_^2−^ concentrations of less than 1000 mg/L (moderate), GBFS or FA can be used to partially replace C. For SO_4_^2−^ concentrations of under 4000 mg/L (severe), only a small quantity of C should be used, and large amounts of GBFS and FA should be added as cementitious materials, preferably in combination with an appropriate amount of CA.

## Figures and Tables

**Figure 1 materials-16-04604-f001:**
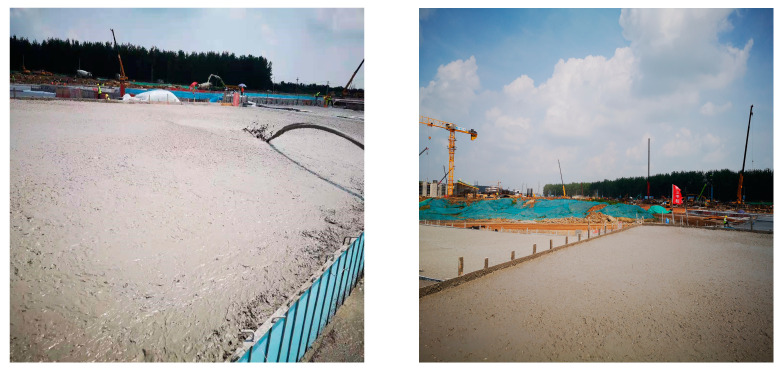
Pouring of FLS for intelligent networked vehicle testing site.

**Figure 2 materials-16-04604-f002:**
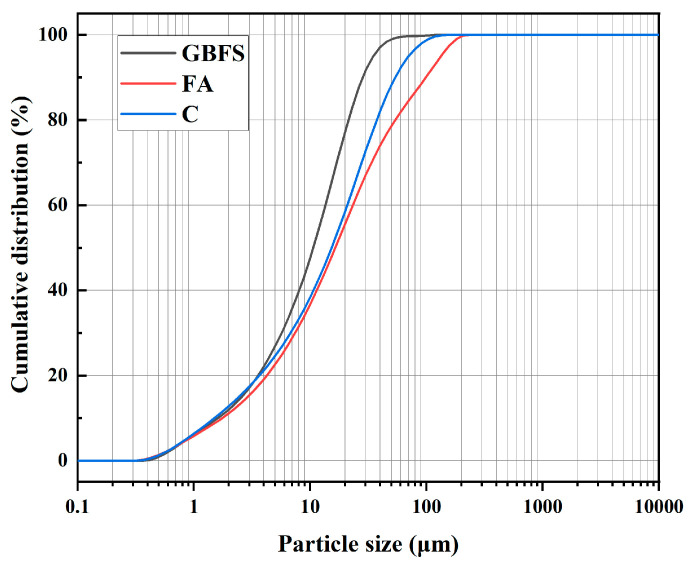
Particle size distribution curves for raw materials.

**Figure 3 materials-16-04604-f003:**
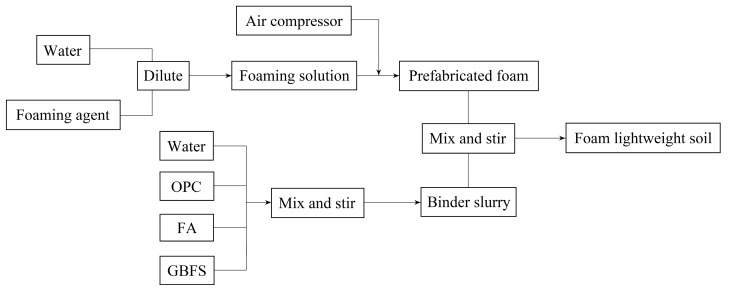
Preparation process of FLS specimens.

**Figure 4 materials-16-04604-f004:**
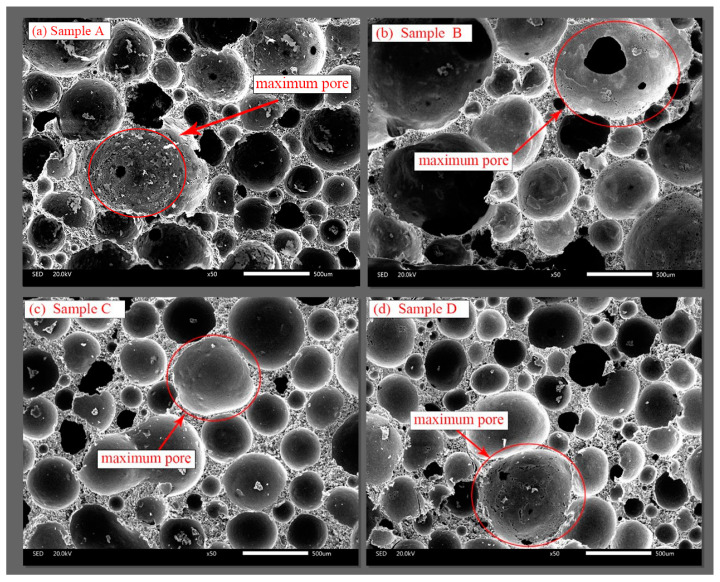
Scanning electron microscopy microstructure of FLS at 56 d: (**a**) Sample A, (**b**) sample B, (**c**) sample C, (**d**) sample D.

**Figure 5 materials-16-04604-f005:**
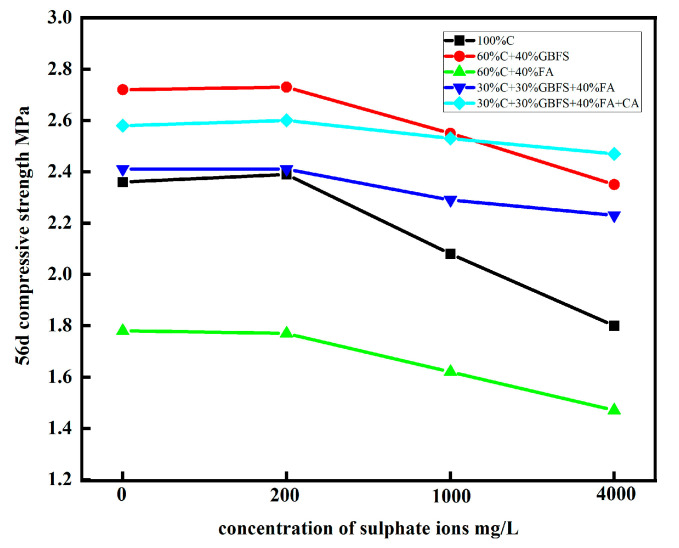
Changes in compressive strengths of specimens eroded by different SO_4_^2−^ concentrations.

**Table 1 materials-16-04604-t001:** Chemical compositions of C, GBFS, FA and CA (wt%).

	CaO	SiO_2_	Al_2_O_3_	Fe_2_O_3_	MgO	SO_3_	K_2_O	Na_2_O	TiO_2_	LOI
Cement	60.11	20.92	5.76	3.24	1.15	2.86	0.88	0.14	0.31	4.17
Granulated Blast Furnace Slag	39.92	31.23	14.12	0.78	7.34	2.23	0.61	0.72	0.76	−0.29
Fly Ash	0.44	57.64	21.49	6.52	1.77	0.37	3.42	0.12	0.93	6.85
Concrete Antiseptic	0.18	58.63	37.91	0.24	0.58	0.07	0.55	0.25	0.04	0.64

**Table 2 materials-16-04604-t002:** Basic properties of P·O 42.5 C.

Density /(kg/m^3^)	Specific Surface Area/(m^2^/kg)	Soundness of Cement/mm	Setting Times/min		Flexural Strength /MPa		Compressive Strength/MPa	
			Initial	Final	3 d	28 d	3 d	28 d
3100	340	2	170	235	5.6	8.7	28.1	50.4

**Table 3 materials-16-04604-t003:** Compositions of FLS specimens (kg/m^3^).

No.	Cementitious Material Systems				Water	Foam
	C	GBFS	FA	CA		
A	345	0	0	0	224	33.2
B	207	138	0	0	224	33.0
C	207	0	138	0	224	32.3
D	103.5	103.5	138	0	224	32.1
E	103.5	103.5	138	5.2	224	31.9

**Table 4 materials-16-04604-t004:** Coefficients of resistance of cementitious materials to sulphate attacks.

No.	A	B	C	D	E
Corrosion resistance coefficient	0.83	0.89	0.87	0.95	0.97

**Table 5 materials-16-04604-t005:** Main performances of FLS.

No.	A	B	C	D	E
Flow factor, mm	170	175	178	175	165
Wet density, kg/m^3^	601	597	596	593	608
7 d Compressive strength, MPa	1.16	0.95	0.63	0.82	0.87
28 d Compressive strength, MPa	2.18	2.12	1.34	1.90	2.05

**Table 6 materials-16-04604-t006:** CO_2_ emissions from cement manufacture in specimens.

No.	A	B	C	D	E
Amount of cement, kg	345	207	207	103.5	103.5
CO_2_ emissions of cement manufacturing, kg	241.5	144.9	144.9	72.5	72.5
CO_2_ reduction rate, %	/	40%	40%	70%	70%

**Table 7 materials-16-04604-t007:** Compressive strength of specimens immersed in solutions with different SO_4_^2−^ concentrations.

No.		A	B	C	D	E
56d compressive strength (contrast specimens), MPa		2.36	2.72	1.78	2.41	2.58
56d compressive strength (experimental specimens), MPa	200 mg/L	2.39	2.73	1.77	2.41	2.60
	1000 mg/L	2.08	2.55	1.62	2.29	2.53
	4000 mg/L	1.80	2.35	1.47	2.23	2.47

## Data Availability

The data that support the findings of this study are available from the corresponding author upon reasonable request.

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
