# Peer review of "Preparation and Properties of Low-Carbon Foamed Lightweight Soil with High Resistance to Sulphate Erosion Environments"

_materials, 2023, doi:10.3390/ma16134604_

Round 1

Reviewer 1 Report

Dear Authors,

please find my comments:

1. Line 48: <been widely built in recent years. They cover large areas and are generally constructed>. What period are Authors talking about?

2. Line 53: <of FLS (Fig. 1). GBFS and FA are usually used to replace C during the preparation of FLS>. Is it necessary to shorten the word Portland cement in the text? 

3. Line 57: <The replacement of ordinary fill with FLS in construction has been extensively studied>. When was it studied - in this study or in previous ones? If in the previous ones, then it should be written that way.

4. Line 70-71: <The effects of FLS composition and microstructure on its properties have been investigated by many researchers as wel>. It is necessary to provide references to other authors.

5. Line 71-80: presented results by other authors, but nothing concrete, i.e. no result values provided?

6. The main question when talking about the introduction - what is the novelty of your research?

7. The introduction does not emphasize what were the shortcomings and what future research needs to be done in this area.

8. The introduction does not emphasize what were the shortcomings and what future research needs to be done in this area.

9. Line 90-96: it would be interesting to know where the raw materials for the research came from.

10. Table 2. Is it really the flexural and compressive strength after 3 days instead of 2 days?

11. Line 174: <The compressive strength of hardened FLS was>. It seems that the sentence is broken, the beginning is there, but the end is not.

12. The authors write that cube-shaped specimens were produced, so why do such names occur - like cube block? Is it correct to call 100 × 100 × 100 mm samples blocks?

13. What equipment was used to perform the tests? Who is the manufacturer? About that needs to be added - In 2.4. section.

14. Figure 3 - can the authors provide a better quality SEM photo? Same note for Figure 4. Graphs of this quality are not recommended for use in a scientific journal. 

Author Response

Thank you for your comments on my article. In accordance with your comments, I have revised the article and responded to your comments. Please see my attachment for details.

Reviewer 2 Report

Dear Sir

the effects of the use of the granulated blast furnace slag, a concrete antiseptic and a fly ash on the compressive strength, sulphate resistance and carbon footprint of foamed lightweight soil were characterized and disused by authors. It is recommended to improve the introduction section by recent references in the last two years. It is recommended to added  XRD patterns of raw materials and for the tested specimens. It is recommended investigate the mechanical properties such as shear strength and flexure strength of the tested specimens

Author Response

(The authors gave the same response as above.)

Reviewer 3 Report

SUMMARY

The article submitted for review is devoted to a topical issue. The issues of preparation and properties of low-carbon foamed lightweight soil with high resistance to sulfate erosion environments have been studied. The relevance is due to the fact that this technology can be applied in large volumes to the road base in the construction of intelligent network test sites for vehicles. There is some scientific deficit in data collection and process and formulation issues on low carbon foam lightweight soils. At the same time, it is important to gain new knowledge about their resistance to sulfate erosion environments. This can be useful not only in scientific terms, but also in applied terms, as all this together leads to the achievement of sustainable development goals to reduce carbon emissions into the environment. Thus, the article has a certain degree of scientific novelty and practical significance. At the same time, there are serious shortcomings in the article, they need to be corrected, they are presented below.

COMMENTS

1.        The authors call their article "Preparation and properties of low carbon foamed lightweight soil ..." However, the issue of preparation is a technological part and is not independent in itself. Probably, the authors should revise the title, adding the word at the beginning "Technology of preparation and properties of low-carbon foamed lightweight soil with high resistance to sulfate erosion environments." Then it will become clear what the purpose of the article was.

2.        The abstract is not well done by the authors. For example, the authors did not formulate a scientific problem at all. It is not clear why the study was conducted and what scientific deficit was eliminated. The authors list the relevance of the work, but do not show what is the scientific deficit and the practical need for the development of such soil. This should be noted in the form of a statement of the research problem at the beginning of the abstract.

3.        Next, the authors provide an introduction in the form of a short literary preface, while the authors reviewed 26 references. However, many of these references are simply listed after the paragraphs and there is no understanding of which works belonged to the authorship of which scientists. This is not a well-executed review and does not reflect the current state of the art. A lot of scientific and engineering works are devoted to the issues of soils and their modification for various purposes. Authors should take the literature review more seriously. From the literature review, the formulation of the scientific deficit should clearly follow, and the goal and objectives of the study should be set. This must be done at the end of section 1.

4.        There is probably a typo in line 86 in the form of a dot before the numbering of section 2. The authors should proofread the article more carefully in terms of editing. There are many technical typos.

5.        Authors must justify the materials selected in subsection 2.1. The process of preparing the mixture is presented in subsections 2.2 and 2.3. Some of the descriptive photographs presented in the sections are of inadequate quality. For example, it is difficult to distinguish the geometric dimensions of the laboratory glassware used in the photo with the ruler shown in Figure 2c. Authors should be more careful with photographic materials.

6.        In addition, attention is drawn to the fact that the preparation of the mixture is stated in the title of the article, however, in the text of the article itself, the preparation of the mixture is given in the Methods section. It should be clearly distinguished whether the developed technology for preparing the mixture is the result of work or is it just one of the methods used. This is important because it can mislead readers.

7.        The results obtained by the authors are interesting, but require additional discussion and comparison with the results of other authors.

8.        The SEM analysis shown in Figure 3 is interesting, but it is of inadequate quality. It has unreadable inscriptions. It is unacceptable. Figure 3 should be submitted in high quality. The same remark applies to Figure 4. The authors did not take seriously the presentation of graphical results. In addition, the number of points along the abscissa in Figure 4 raises questions. From the fact that 4 points are presented, the figures look somewhat uninformative, while the selected ranges also raise questions. For example, between the marks 200 and 1000 there is exactly the same interval as between 1000 and 4000. That is, the authors must clearly consider the scale of their drawing. To date, the discussion of the results obtained does not reflect the stated goal.

9.        The conclusions should be supplemented in terms of new scientific knowledge obtained during the study, with a clear formulation of recommendations for applying the results of the study and reflecting the prospects for the development of scientific research in new articles.

10.     26 references analyzed by the authors need to be improved. First, the number of analyzed sources does not reflect scientific novelty. The list of references should be supplemented with at least 35 references. Secondly, attention is drawn to a large number of obsolete materials that are older than the last 5 years. Soil technologies, strengthening and fixing of soils are developing every year and there are quite a lot of articles on this topic in modern journals. This is very important because it reflects the relevance and novelty of the research.

11.     In general, the article also needs editorial proofreading, because its style of presentation does not yet correspond to articles of a level that can be accepted for a journal.

Unfortunately, while the reviewer does not see the opportunity to publish this article in the specified form, serious improvements are needed on all the comments made. The reviewer invites the authors to finalize the article, making serious changes and send it for re-reviewing. General conclusion of the reviewer: major revisions.

Minor corrections are required in the entire text of the manuscript.

Author Response

(The authors gave the same response as above.)

Reviewer 4 Report

In this study the authors have prepared the lightweight foamed soil using different combination of Portland cement (C), granulated blast furnace slag (GBFS), fly ash (FA) and a small amount of a concrete antiseptic. Very limited experiments were performed. The following comments should consider in revising the manuscript.

In the abstract it is mentioned that the FLS preparation had CO2 emission reduction rate of up to 70%. How it was measured as in the manuscript there is nothing about CO2 emission.

What references/standards were followed in calculating the Corrosion resistance coefficients? Please mention it.

In page 6, line 178-182: Please mention the value of SO42- concentrations and appropriate age of maintenance

Figure 4 quality is very poor. It must be improved for better readability.

A cost analysis should be included for FLS preparation and compare it with conventional soils typically used.

English language is OK.

Author Response

(The authors gave the same response as above.)

Round 2

Reviewer 1 Report

I recommend the Authors to change the title of the Table 2 (Table 2 Physical properties of P·O 42.5 C) and Table 5 Physical properties of FLS because properties such as Flexural strength and Compressive strength - are not a physical properties.  

Author Response

Dear reviewer,

Thank you again for your valuable recommendations on the manuscript.

We have changed the tile of the Table 2 to Basic properties of P·O 42.5 C.

We have changed the tile of the Table 5 to Performances of FLS.

Reviewer 3 Report

The authors have improved the quality of the article and reasonably responded to the comments. The reviewer believes that the article can be published in its current form.

Author Response

Dear reviewer,

Thank you again for your valuable advice on the paper.

The comments you made have improved the quality of our manuscript.

Reviewer 4 Report

Just a minor issue. In Fig 2, Y-axis, will it be cumulative volume (%) or cumulative weight (%)? Typically, we do sieve analysis based on weight of the materials.

English is acceptable

Author Response

Dear reviewer,

Thank you again for your advice on the manuscript.

Reviewing relevant literatures, we have modified the Y-axis as Cumulative distribution (%) to ensure that there is no misunderstanding.